# Development of a Cost-Effective Process for the Heterologous Production of SARS-CoV-2 Spike Receptor Binding Domain Using *Pichia pastoris* in Stirred-Tank Bioreactor

Diego G. Noseda [1,2], Cecilia D'Alessio [1,3,4], Javier Santos [1,3,4], Tommy Idrovo-Hidalgo [3], Florencia Pignataro [1,3], Diana E. Wetzler [1,4,5], Hernán Gentili [3], Alejandro D. Nadra [1,3], Ernesto Roman [1,4], Carlos Paván [1,6] and Lucas A. M. Ruberto [7,8,9,*]

1   Consejo Nacional de Investigaciones Científicas y Técnicas (CONICET),
    Godoy Cruz 2290, Buenos Aires C1425FQB, Argentina; dnoseda@iib.unsam.edu.ar (D.G.N.)
2   Instituto de Investigaciones Biotecnológicas (IIBio), Universidad Nacional de San Martín-CONICET,
    Av. 25 de Mayo 1301-1367, Villa Lynch, Provincia de Buenos Aires 1650, Argentina
3   Facultad de Ciencias Exactas y Naturales, Departamento de Fisiología y Biología Molecular y Celular,
    Instituto de Biociencias, Biotecnología y Biología Traslacional (iB3), Ciudad Universitaria-Pabellón II,
    Buenos Aires 1428, Argentina
4   Departamento de Química Biológica, Facultad de Ciencias Exactas y Naturales, Universidad de Buenos Aires,
    Ciudad Universitaria-Pabellón II, Ciudad Autónoma de Buenos Aires 1428, Argentina
5   Instituto de Química Biológica de la Facultad de Ciencias Exactas y Naturales (IQUIBICEN),
    CONICET-Universidad de Buenos Aires, Ciudad Universitaria-Pabellón II,
    Ciudad Autónoma de Buenos Aires 1428, Argentina
6   Instituto de Química y Fisicoquímica Biológicas, LANAIS PROEM, Facultad de Farmacia y Bioquímica,
    Universidad de Buenos Aires, Ciudad Autónoma de Buenos Aires C1113AAD, Argentina
7   Departamento deMicrobiología, Inmunología, Biotecnología y Genética, Facultad de Farmacia y Bioquímica,
    Universidad de Buenos Aires, Junín 956 6to Piso, Ciudad Autónoma de Buenos Aires C1113AAD, Argentina
8   Facultad de Farmacia y Bioquímica, Instituto de Nanobiotecnología (NANOBIOTEC), CONICET-Universidad
    de Buenos Aires, Junín 956 6th Floor, Ciudad Autónoma de Buenos Aires C1113AAD, Argentina
9   Instituto Antártico Argentino, Ministerio de Relaciones Exteriores y Culto, Buenos Aires, Argentina.
    Av. 25 de Mayo 1147, Villa Lynch, Provincia de Buenos Aires 1650, Argentina
*   Correspondence: luruberto@gmail.com

**Abstract:** SARS-CoV-2 was identified as the pathogenic agent causing the COVID-19 pandemic. Among the proteins codified by this virus, the Spike protein is one of the most-external and -exposed. A fragment of the Spike protein, named the receptor binding domain (RBD), interacts with the ACE2 receptors of human cells, allowing the entrance of the viruses. RBD has been proposed as an interesting protein for the development of diagnosis tools, treatment, and prevention of the disease. In this work, a method for recombinant RBD production using *Pichia pastoris* as a cell factory in a stirred-tank bioreactor (SRTB) up to 7 L was developed. Using a basal saline medium with glycerol, methanol, and compressed air in a four-stage procedure, around 500 mg/L of the raw RBD produced by yeasts (yRBD) and 206 mg/L of purified (>95%) RBD were obtained. Thereby, the proposed method represents a feasible, simple, scalable, and inexpensive procedure for the obtention of RBD for diagnosis kits and vaccines' formulation.

**Keywords:** SARS-CoV-2; RBD; antigen; STBR; *Pichia pastoris*





## 1. Introduction

The 2020, the SARS-CoV-2 pandemic demanded the development of suitable tools to face and manage a widespread human infection. Although massive contagion seems to be currently a picture of the recent past, novel viral variants and global increases in the number of infected people, such as those taking place in China and the Northern Hemisphere during the second half of 2022, are issues of current concern related to SARS-CoV-2 [1].

Viral proteins or some of its domains would be useful molecules to detect, treat, and prevent viral disease and its consequences. SARS-CoV-2 belongs to the Coronaviridae family [2,3]. Coronaviruses are enveloped non-segmented positive-sense RNA viruses. SARS-CoV-2 virus presents a genome with four open reading frames (ORFs) for the structural proteins: Spike, Envelope, Membrane, and Nucleocapsid. The Spike complex (~150 kDa) mediates the viral and cellular membrane interaction and fusion by binding mainly to angiotensin-converting enzyme 2 (ACE2) through the receptor binding domain (RBD) [4–7]. Some regions of the Spike protein were suggested as suitable targets for drug development [8]. Considering this specific function, it is possible to assume that RBD heterologous expression would provide a useful tool for diagnosis purposes, as well as for immunization to obtain neutralizing antibodies or even a protein-based vaccine. It was reported that in human natural infection, a large fraction of the neutralizing antibodies target RBD [9–11]. What is more, Liu et al. (2020), working with nineteen potent neutralizing antibodies (in vitro) obtained from infected patients, found that almost half of them were directed against the RBD [10], highlighting its potential role as a vaccine antigen. For these reasons, SARS-CoV-2 RBD was selected for its heterologous expression aiming to obtain large amounts of such protein.

The heterologous production of several SARS-CoV-2 proteins was reported using different expression systems, the whole Spike protein and its RBD being the most-common ones [11–14]. For example, Li et al. (2020) [15] expressed the RBD, the S1 subunit, the WT S ectodomain, and a prefusion-trimer-stabilized form of S using Sf9 insect cells.

It is important to consider that the Spike protein presents 22 possible N-glycosylations and 4 O-glycosylation sites, some of them being on the RBD [16,17]. Additionally, the RBD presents 9 cysteines, 8 of them forming S-S bridges [8]. Glycosylation, as well as disulfide bridges are issues of special attention for heterologous protein production, since these kinds of modifications affect protein folding and, in some cases, biological activity [18]. This is particularly relevant when the recombinant proteins are produced for medical use in humans [19,20]. The selection of the expression system is usually strongly conditioned by the requirement of such post-translational modifications [21]. In the case of the RBD, glycosylations and S-S bridges' formation seem to be important for adequate protein folding, and thus, the expression host selection is a critical decision in the development of its production process [22].

Some yeasts such as *Saccharomyces cerevisiae*, *Kluyveromyces lactis*, *Yarrowia lipolytica*, and *Pichia pastoris* (*Komagataella phaffii*) are suitable and convenient hosts for recombinant protein production [23]. Singularly, *P. pastoris* is a methylotrophic non-conventional yeast considered as a biological model [24,25] and used for heterologous protein production, usually taking advantage of the strong alcohol oxidase 1 (AOX1) promoter and its ability to achieve high cellular density in bioreactors, for which values near 100 g dry cell weight (DCW)/L have been reported [26–28]. The AOX1 promoter strongly responds to methanol, while its activity is repressed by glucose and glycerol [29,30]. This yeast is also able to secrete large amounts of properly folded heterologous proteins with only a few other secreted proteins and is, for these reasons, widely used as an expression system. Additionally, this microorganism is able to perform some post-translational protein modifications (glycosylation, proteolytic processing, and disulfide bonds formation) usually observed in higher eukaryotes, a relevant feature when the production of proteins for medical purposes is involved [31–34] or when those modifications are required for proper protein folding [35–37]. Beyond heterologous protein production, *P. pastoris* has been recently used for the expression of other metabolic pathways, leading to the obtention of non-proteinous molecules [38]. It has also been evaluated as a key component in a probiotic preparation for poultry [39]. Finally, a complete reference genome of these microorganisms is available [40].

Optimum conditions for heterologous protein production using *P. pastoris* depend on several factors, such as the medium's composition, temperature, and culture strategy, among others [41,42]. For this reason, research must be performed to find the conditions that maximize protein production with the selected microbial construction [43]. Methanol

feeding strategies, flows, and concentration are considered as relevant factors affecting cellular activity and protein production [43,44]. Staggered fed, exponentially fed, DO-stat, and methanol concentration feedback control are strategies commonly reported for *P. pastoris* cultivation [45]. Among them, DO-stat consists of looping the methanol feeding to the dissolved oxygen concentration. In such a way, methanol feeding is activated as pulses when the percent of dissolved oxygen rises beyond a setpoint, thus avoiding an excessive $O_2$ demand, heat production, as well as anaerobiosis or methanol accumulation [46–48].

Antigens such as recombinant RBD are considered useful for subunit vaccines' development, especially when they are produced using high-yield hosts such as *P. pastoris*, allowing the production of large amounts of antigen doses at a relatively low cost [12,13,49], for this reason and its scalability are being a suitable tool to face a pandemic.

The SARS-CoV-2 pandemic presented nucleic-acid-based vaccines as state-of-the-art tools for massive vaccination; however, some safety and logistical aspects of this recently implemented vaccination strategy for humans raised concerns among a great part of the population. The toxicity of synthetic raw materials used to conjugate lipids in mRNA vaccines, the possibility of nucleic acids persisting in vivo, and the risk of the theoretical integration of foreign DNA into the host chromosome [50], in addition to the high costs and strict cold chain requirements for some vaccines based on mRNA—having a difficult distribution in remote areas where ultra-low temperature freezers are unavailable—represent some of its disadvantages. Although this new technology is promising, protein subunit vaccines are also a functional and safe alternative. Due to their higher safety profile, subunit vaccines are primarily developed for use with elderly and infant patients [51,52], and vaccines based on the RBD alone effectively boost an immune response originally generated against a full-length Spike protein trimer, increasing interest in using RBD-based vaccine boosters to provide immunity against emerging variants [53]. Additionally, protein subunit vaccines do not require ultra-freezing conditions and can also be safely stored in a regular fridge or lyophilized for their distribution [54], making this type of vaccine useful for complementing vaccination campaigns all over the world.

In a previous work, we reported that recombinant SARS-CoV-2 RBD produced using *P. pastoris* as the expression host presented a similar and comparable conformation as the one produced using HEK293T mammalian cells. A bioreactor production procedure was used, yielding 45 mg/L of 90% pure protein [13]. This was a first attempt at the production of the RBD at a scale large enough for small-scale protein characterization and immunization assays.

In this work, we propose a new procedure that improves by five-times the production yield of the recombinant RBD antigen from the SARS-CoV-2 Spike protein. It consists of a four-step procedure that was optimized by comparing two culture strategies in a 7 L stirred-tank bioreactor. Furthermore, we report the scaling up of the procedure of RBD production to a 14 L stirred-tank bioreactor.

Goal: The goal of the project was to produce a low-cost antigen to be used in diagnosis (antibodies detection), therapies (generation of neutralizing antibodies), and prevention (vaccine antigen production).

## 2. Methods

### 2.1. Plasmid and Strain

The RBD sequence and plasmid construction, as well as the *Pichia pastoris* strain used in this work were the same as described previously [12,13]. Briefly, the sequence of amino acid residues 319–537 of the SARS-CoV2 Spike protein (RBD) was codon-optimized for expression in *P. pastoris*. Furthermore, the alpha-factor secretion signal (N-terminal) of *S. cerevisiae* was fused to direct the heterologous protein into the culture medium. In addition, a *Staphylococcus aureus* sortase A recognition sequence for covalent coupling and a His6 tag for purification were inserted into the C-terminal extreme. The entire sequence was synthesized and cloned into the pPICZalphaA vector under transcriptional control

of the *P. pastoris* AOX1 promoter by GenScript (Piscataway, NJ, USA) and linearized and transformed into the *P. pastoris* X-33 strain.

### 2.2. Determination of Dry Cell Weight

The optical density of *P. pastoris* samples from the shake flask cultures and bioreactor fermentations was determined at 600 nm using a UV–Vis spectrophotometer and converted to dry cell weight (DCW, in g/L) using the following equation: $DCW = 0.269 \, OD_{600nm}$ ($R^2 = 0.99$), corresponding to a calculated DCW to $OD_{600nm}$ calibration curve.

### 2.3. Quantification of Total Proteins and RBD

Samples collected during the methanol induction phase of the flask cultures and bioreactor fermentations were centrifuged at $10,000\times g$ for 20 min to obtain cell-free supernatants. The total protein concentration during cultures and fermentations was determined in the supernatants by the Bradford method [55,56] using a calibration curve of the BSA standard. Proteins in the supernatants were run in a 12% SDS-PAGE stained with Coomassie brilliant blue G-250 (Sigma-Aldrich; St. Louis, MO, USA) to visualize protein bands. Protein quantification was performed comparing densitometry with known standards of BSA. Chen et al. (2017) [57] reported that Coommasie Blue dye binds 1.33-fold less to RBD from SARS-CoV than to BSA. It is worth mentioning that, although in this work, we did not compare the binding capability of RBD from SARS-CoV2 to Coommasie Blue with that of BSA and, thus, there may be some underestimation of the yield of the process, this will be the same in all samples. The relative abundance of recombinant RBD, in both supernatants and purification fractions, was determined by band densitometry using the ImageJ software (http://rsb.info.nih.gov/ij, accessed on 10 December 2020) and considering the total protein concentration determined by the Bradford method.

### 2.4. Medium Composition for Flask Cultures and Bioreactor Fermentations

The cultivation of *P. pastoris* inoculums in Erlenmeyer flasks was performed either in BMGY medium (1% yeast extract, 2% bactopeptone, 1.34% YNB, 400 µg/L biotin, 100 mM potassium phosphate pH 6, and 1% glycerol) or in a low-salt medium (LSM) with 10 g/L glycerol at 30 °C and 250 rpm. The LSM medium contained: 4.55 g/L potassium sulfate, 3.73 g/L magnesium sulfate heptahydrate, 1.03 g/L potassium hydroxide, 0.23 g/L calcium sulfate anhydrous, and 10.9 mL/L phosphoric acid 85%. After the sterilization of the medium, 3.5 mL/L of filtered biotin solution (0.2 g/L) and 3.5 mL/L of filtered trace metal solution (PTM1) were added. PTM1 contained per liter: 6.0 g copper (II) sulfate pentahydrate, 0.08 g sodium iodide, 3.0 g manganese sulfate-monohydrate, 0.2 g sodium molybdate-dihydrate, 0.02 g boric acid, 0.5 g cobalt chloride, 20.0 g zinc chloride, 65.0 g ferrous sulfate-heptahydrate, 0.2 g biotin, and 5.0 mL sulfuric acid (98% *w/w*). The cultivations of *P. pastoris* in the bioreactors were performed in LSM supplemented with 40 g/L glycerol. The use of LSM prevents salt precipitation during the pH rise in the downstream process, as was previously reported [57].

### 2.5. Inoculum Preparation

To obtain the inoculum for flask cultures and bioreactor fermentations, a single colony of *P. pastoris* clone grown on a YPD agar plate was inoculated into a 250 mL flask containing 40 mL of LSM (supplemented with PTM1 and biotin) with 10 g/L glycerol or in 250 mL BMGY and cultured overnight at $30 \pm 1$ °C and 250 rpm in an orbital shaker. A volume of 400 mL of LSM (supplemented with PTM1 and biotin) containing 10 g/L glycerol in a 2 L Erlenmeyer flask was inoculated with the overnight culture and incubated under the same conditions until the culture reached an $OD_{600}$ of ~14. This culture was used to simultaneously inoculate a set of Erlenmeyer flasks and a stirred-tank bioreactor with LSM at a ratio of $V_{seed} = V_0/10$, where $V_{seed}$ is the volume of the inoculum and $V_0$ is the initial volume of the culture.

### 2.6. Cultivation in Erlenmeyer Flask

Cultures of *P. pastoris* expressing the RBD clone were started in Erlenmeyer flasks to evaluate the growth kinetics and recombinant RBD production. For this purpose, three 250 mL Erlenmeyer flasks containing 50 mL of LSM (supplemented with PTM1 and biotin) with 10 g/L glycerol were inoculated with 5 mL of the previously described culture and incubated at $30 \pm 1$ °C and 250 rpm in an orbital shaker. After 24 h, the induction conditions were established by adding pure methanol at a final concentration of 1% ($v/v$). This procedure was repeated every 24 h to maintain methanol induction for a period of 120 h. During induction, flasks were incubated on a shaker at 250 rpm and $25 \pm 1$ °C, as low culture temperatures increase the yield of soluble recombinant proteins in *P. pastoris* due to reduced extracellular proteolysis without affecting cell growth [58,59]. Culture samples for the determination of biomass, total proteins, and RBD concentration were collected every 24 h. Specific growth rates (μ) for each culture stage were calculated from the slope of the regression line of the growth curve.

### 2.7. Fermentations in Stirred-Tank Bioreactor

Fermentations were carried out in a stirred-tank bioreactor (BioFlo 115, New Brunswick Scientific; Edison, NJ, USA) using a four-stage procedure based on previous work with modifications [13,60–62]. The first stage consisted of a batch culture using LSM with unlimited glycerol (40 g/L) as the sole carbon and energy source, supplemented with 3.5 mL/L PTM1 and 3.5 mL/L biotin solution (0.2 g/L). After an abrupt peak in the percentage of dissolved oxygen (Spike) indicating carbon source depletion, the second phase—fed-batch with glycerol—was initiated. In this phase, the culture was fed with a solution containing 600 g/L of glycerol, 12.25 mL/L PTM1, and 12.25 mL/L biotin solution (0.2 g/L). To ensure glycerol limitation and, thus, gradually derepress the AOX1 promoter, feeding was automatically regulated according to the percentage of dissolved oxygen (%DO) in the culture, a strategy referred to as DO-stat [46,63]. Later, a short transition phase was performed to allow the adaptation of the culture to growth in the presence of methanol as the sole carbon source. For this, two strategies were compared: (1) feeding with a glycerol/methanol mixture (3:1) for 5 h and (2) feeding with a pulse of 4 g/L of methanol. Finally, the last induction phase was initiated by adding pure methanol, supplemented with 12.25 mL/L PTM1 and 12.25 mL/L biotin solution (0.2 g/L), as the sole carbon and energy source, applying a fed-batch procedure with a growth-limiting rate. The feeding of methanol at limiting concentration was also regulated with the level of DO in the culture (DO-stat). The stirred-tank bioreactor operated by the interface of the Biocommand Bioprocessing (New Brunswick Scientific) software for parameter control and data acquisition. The temperature was maintained at $30 \pm 1$ °C during the batch and glycerol-fed-batch phases and at $25 \pm 1$ °C during the transition and induction phases. The pH was kept at 5.0 in the first two phases and 5.3 in the last two phases by adding 42.5% ($v/v$) $H_3PO_4$ and 14% ($v/v$) $NH_4OH$, which also served as a nitrogen source. Dissolved oxygen was controlled by stirring (maximum 1000 rpm) and by filtered (0.22 μm) compressed air (1–1.5 VVM). The pH was measured using a pH electrode (Mettler-Toledo GmbH, Gießen, Hessen, Germany), and the dissolved oxygen content was determined using a polarographic probe (InPro6110/320, Mettler-Toledo GmbH). Foam formation was prevented by adding 3% ($v/v$) antifoam agent 204 (Sigma-Aldrich; St. Louis, MO, USA). Samples were taken during the different fermentation phases to determine the concentrations of the biomass, total protein, and recombinant RBD. Biomass evolution during the fermentation was expressed as DCW (g/L) = f(t). Fermentations were performed in vessels of 7 and 14 L to compare the RBD production process with two different vessel volumes. After fermentation, the biomass was removed from the culture by centrifugation at $18,600 \times g$ for 20 min at 4 °C in a Sorvall high-speed centrifuge (Lynx 4000 Thermo, Waltham, MA, USA) equipped with a F10 rotor, and the supernatant was used for RBD purification.

### 2.8. Purification of Recombinant RBD and Quality Control

The purification of recombinant RBD from cell-free supernatants, as well as RBD quality control was as already described [13]. Briefly, the purification was performed using a NTA-Ni$^{2+}$ column, previously equilibrated with 20 mM Tris–HCl, 150 mM NaCl, and 20 mM imidazole, pH 7.4 (equilibration solution). The supernatants were adjusted to pH 7.4 with NaOH and to 20 mM Tris and 20 mM imidazole, centrifuged 20 min at 12,000× *g*, and loaded into the column. The column was washed with an equilibration solution. Finally, the recombinant RBD was eluted in Tris 20 mM, NaCl 150 mM, and 300 mM imidazole, pH 7.4. The purified protein was dialyzed twice in 20 mM Tris–HCl and 150 mM NaCl buffer, pH 7.4. Absorption spectra (240–340 nm range, using a 0.1 nm sampling interval) were acquired at 20°C with a JASCO V730 BIO spectrophotometer (Japan). The RBD concentration was determined using: $\varepsilon$ = 33,850 M$^{-1}$ cm$^{-1}$ (Abs = 1.304 for a 1 mg mL$^{-1}$ protein solution). After that, the RBD was stored at −80°C. The purity of the recombinant RBD was analyzed by reverse-phase HPLC using an analytical C18 column (Higgins Analytical, Inc., Mountain View, CA, USA) and a JASCO system (equipped with an autoinjector, an oven at 25 °C, and a UV detector). The elution was made at a 1.0 mL/min flow using an ACN gradient from 0 to 100% over 40 min (10–50 min of the run) with a mobile phase of 0.05% TFA.

### 2.9. Statistical Analysis

The data for the cell density, total protein concentration, and RBD concentration were taken in triplicate. The means and standard deviations were calculated. Statistical significance was evaluated by the Student's *t*-test using the Infostat V2020 Software. Differences were considered significant if $p < 0.05$.

## 3. Results

### 3.1. Growth Kinetics and RBD Expression at Flask Level

The *P. pastoris* clone expressing RBD [13] was cultured in Erlenmeyer flasks to evaluate the growth kinetics and recombinant RBD expression. The culture in LSM containing 10 g/L glycerol exhibited a maximum specific growth rate ($\mu_{max}$) of 0.15 h$^{-1}$ and a biomass yield coefficient based on consumed substrate ($Y_{X/S}$) of 0.43 g DCW/g glycerol (Figure 1A). At the end of the exponential phase, corresponding to an incubation period of 24 h, the culture displayed a cell concentration of 4.6 g DCW/L, while the glycerol was completely depleted. Moreover, during the expression induction phase, in which methanol was added in pulses, cells of the *P. pastoris* clone continued growing with a specific growth rate ($\mu$) of 0.072 h$^{-1}$ in the first 24 h of induction. The culture continued growing, showing a decrease in the growth rate, due to methanol limitation, to a final value of 0.002 h$^{-1}$. Hence, the average specific growth rate over the induction phase was 0.026 h$^{-1}$. After 120 h of methanol induction, the culture reached a maximum biomass level of 8.0 g DCW/L in the induction phase with a biomass yield coefficient ($Y_{X/S}$) of 0.085 g DCW/g methanol on average (Figure 1A). Figure 1B shows the decrease in the specific growth rate during the methanol induction phase.

As shown in Table 1, the total protein concentration in the culture supernatant increased from 5.2 mg/L at the beginning of the methanol induction phase to 72.3 mg/L after 120 h of induction. Moreover, the RBD concentration was 1.6 mg/L at the beginning of this phase and 21.7 mg/L at the end, representing a 14-fold increase. The percentage of the recombinant RBD in the supernatants with respect to the concentration of total proteins was 30%. The RBD yield based on biomass formation ($Y_{RBD/X}$) showed a value of 2.71 mg/g at 120 h methanol induction, corresponding to an eight-fold increase with respect to the initial value. Furthermore, the volumetric RBD productivity (vP) at the end of incubation was 0.15 mg/Lh, representing an increase of 100% compared to the initial induction time (Table 1). The specific RBD productivity (sP) was 18.8 µg RBD/g DCW h at the end, indicating that this parameter increased 1.4-fold with respect to the initial point (Table 1). Finally, the RBD yield based on methanol consumed ($Y_{RBD/S}$) reached

0.54 mg RBD/g methanol at the end of the whole process. It is worth mentioning that the results obtained with Erlenmeyer cultures were important for planning the scale-up of the recombinant RBD production in the stirred-tank bioreactor.

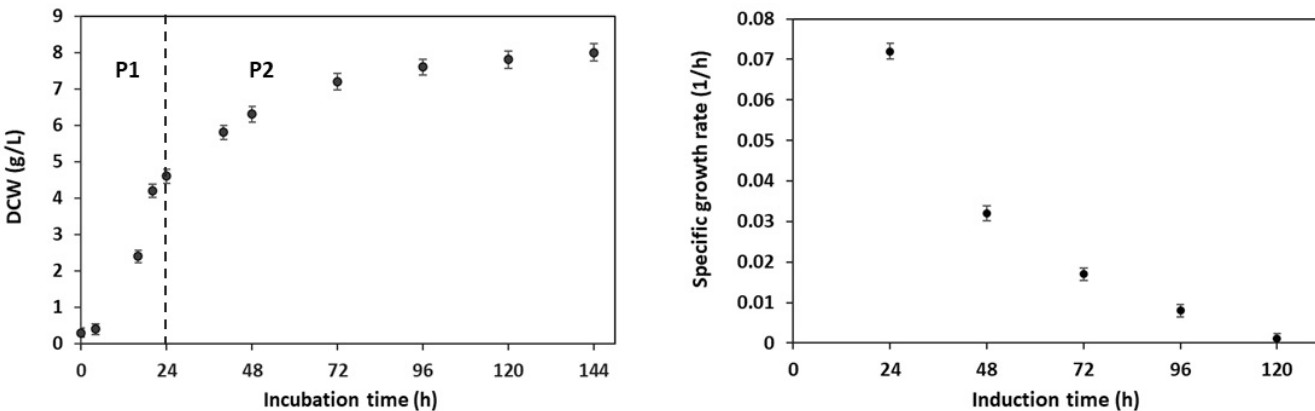

**Figure 1. Growth of *P. pastoris* expressing RBD in flask cultures.** (**A**) Biomass concentration (g DCW/L) evolution during flask cultivation. **P1**: exponential growth phase in LSM with 10 g/L glycerol, **P2**: Induction phase with methanol pulses at a final concentration of 1% (*v/v*), each 24 h incubation. Error bars indicate 2 SD. (**B**) Variation of the specific growth rate ($\mu$) during the induction phase. During the induction of expression, the culture continued growing with a decrease in the growth rate due to the limitation of methanol.

**Table 1.** Parameters obtained from the culture of *P. pastoris* expressing RBD in Erlenmeyer flasks. Detailed parameters: Total protein concentration (mg/L), RBD concentration (mg/L), RBD concentration increase (fold), $Y_{RBD/X}$ (mg RBD/g DCW): RBD yield based on biomass formation; $_V$P (mg RBD/L h): volumetric RBD productivity, $_S$P (ug RBD/g DCW h): specific RBD productivity.

| Induction Time (h) | Total Protein Concentration (mg/L) | RBD Concentration (mg/L) | RBD Increase (fold) | $Y_{RBD}/x$ (mg/g) | vP (mg/L h) | sP ($\mu$g/g h) |
|---|---|---|---|---|---|---|
| 0 | 5.2 | 1.6 | 1.0 | 0.34 | 0.07 | 14.1 |
| 24 | 12.8 | 3.8 | 2.5 | 0.61 | 0.08 | 12.7 |
| 48 | 26.4 | 7.9 | 5.7 | 1.12 | 0.11 | 15.5 |
| 72 | 41.8 | 12.5 | 8.0 | 1.67 | 0.13 | 17.4 |
| 96 | 57.1 | 17.1 | 11.0 | 2.17 | 0.14 | 18.1 |
| 120 | 72.3 | 21.7 | 13.9 | 2.71 | 0.15 | 18.8 |

### 3.2. Production of Recombinant RBD in 7 L and 14 L Stirred Bioreactors

The fermentation of the *P. pastoris* clone for the production of the recombinant RBD was first performed in a 7 L stirred-tank bioreactor according to the four-phase procedure described above. Two fermentation strategies were compared, using an initial culture volume of 1.5 L in the bioreactor in both cases. The fermentation carried out with Strategy 1 showed a maximum biomass concentration of 16.7 g DCW/L at 18 h of the batch phase. During this stage, the *P. pastoris* culture displayed a maximum specific growth rate ($\mu_{max}$) of 0.16 h$^{-1}$ and a biomass yield coefficient ($Y_{X/S}$) of 0.42 DCW/g glycerol. After an increase in dissolved oxygen due to glycerol depletion (DO spike), the fed-batch phase was initiated, with glycerol feeding controlled by the percentage of dissolved oxygen (%DO), with a cut-off of 50% saturation. Glycerol feeding was maintained for 23 h, reaching a biomass level of 61.4 g DCW/L. Next, the transition phase was performed by feeding the culture with a glycerol:methanol (3:1) mixture for 5 h, allowing the cells to adapt to the methanol and reaching a cell concentration of 64.1 g DCW/L at the end of this phase. Then, the methanol-fed-batch phase was carried out to induce the expression of the recombinant

RBD, regulating the feeding of pure methanol in response to %DO with a saturation cut-off of 60%. After 52 h of methanol induction and a total fermentation process of 98 h, the culture reached a biomass level of 78.2 g DCW/L. At this fermentation time, the total protein concentration in the culture supernatant was 296.3 mg/L, while the RBD reached 98.4 mg/L, corresponding to 33% of the total proteins in the supernatant (Table 2). Since the volume of the supernatant was 2.6 L, a total amount of the RBD of 255.8 mg was obtained. The RBD yield based on biomass formation ($Y_{RBD/X}$) exhibited a value of 1.3 mg RBD/g DCW at the end of the induction. The volumetric RBD productivity ($_{v}P$) of the whole process reached 1.0 mg RBD/L h, and the total RBD productivity ($_{T}P$) was 2.6 mg RBD/h (Table 2). Moreover, the specific RBD productivity ($_{S}P$) of the whole process was 12.8 µg RBD/g DCW h, and the RBD yield based on methanol consumed ($Y_{RBD/S}$) reached 0.5 mg DCW/g methanol at the end of the fermentation.

**Table 2. Production parameters of *P. pastoris* fermentations using Strategies 1 and 2 in 7 L stirred-tank bioreactor.** Detailed production parameters: Final biomass level (g DCW/L); total protein concentration (mg/L); RBD concentration (mg/L); total RBD (mg); $Y_{RBD/biomass}$ (RBD yield based on biomass formation, mg/g); whole process volumetric RBD productivity ($_{v}P$, mg RBD/L h); whole process total RBD productivity ($_{T}P$, mg RBD/h); whole process specific RBD productivity ($_{S}P$, ug RBD/g DCW h); $Y_{RBD/Methanol}$ (final RBD yield based on methanol consumed, mg/g).

|  | Strategy 1 | Strategy 2 |
|---|---|---|
| Final biomass level (g DCW/L) | 78.2 | 89.2 |
| Total protein concentration (mg/L) | 296.3 | 1378.5 |
| RBD concentration (mg/L) | 98.4 | 519.6 |
| Total RBD (mg) | 255.8 | 1402.9 |
| YRBD/Biomass (mg/g) | 1.3 | 5.8 |
| Volumetric RBD productivity (mg/L h) | 1.0 | 3.8 |
| Total RBD productivity (mg/h) | 2.6 | 10.3 |
| Specific RBD productivity (µg/g DCW h) | 12.8 | 42.2 |
| YRBD/Methanol (mg/g) | 0.5 | 1.8 |

The fermentation performed with Strategy 2 in the 7 L stirred-tank bioreactor presented a maximum biomass level of 17.2 g DCW/L at the end of the batch phase (18 h). At this stage, the culture showed similar values of the specific growth rate and biomass yield coefficient as in Strategy 1. In the fed-batch phase, feeding with glycerol was maintained for 8 h in response to %DO with a saturation cut-off of 50%, reaching a biomass level of 40.9 g DCW/L. Then, in the transition phase, feeding was carried out with a dose of 4 g/L methanol, which allowed culture adaptation to methanol and to reach a biomass concentration of 41.4 g DCW/L after 4 h. Complete consumption of methanol was indicated by an abrupt increase of %DO (DO spike). During the induction phase, methanol feeding was regulated by %DO with a saturation cut-off of 60%, reaching a final biomass level of 89.2 g DCW/L after 108 h of induction and 138 h of total process. At this time, the total protein and RBD concentrations in the culture reached 1378.5 mg/L and 519.6 mg/L, respectively, while the total recombinant RBD yielded 1402.9 mg, as the supernatant volume was 2.7 L (Table 2). Thereby, the RBD yield based on biomass ($Y_{RBD/X}$) at the end of the process showed a value of 5.8 mg RBD/g DCW, an increase of 4.5-fold compared to Strategy 1. The volumetric RBD productivity ($_{v}P$) of the whole process was 3.8 mg RBD/L h, and the total RBD productivity ($_{T}P$) reached 10.3 mg RBD/h (Table 2), which corresponds to an approximately 4-fold increase of both parameters compared to Strategy 1. Moreover, the specific RBD productivity ($_{S}P$) of the whole process was 42.2 µg RBD/g DCW h, and the RBD yield based on methanol consumed ($Y_{RBD/S}$) reached 1.8 mg RBD/g methanol at the end of the fermentation, corresponding to an approximately 3.5-fold

increase of both parameters compared to Strategy 1. Figure 2 shows the parameter profile of the fermentation carried out with Strategy 2 and describes the variation of the stirring, temperature, pH, dissolved oxygen, and feeding throughout the fermentation process. This profile reports the decrease in dissolved oxygen content due to culture growth during the batch phase until the dissolved oxygen spike, the point at which glycerol feeding began, and also shows the variations in dissolved oxygen content by which glycerol or methanol feeding was regulated. SDS-PAGE analysis of the fermentation supernatants corresponding to Strategy 2 displayed the increase in the total protein and recombinant RBD during the expression induction phase with methanol. The three bands of around 30, 35, and 40 kDa correspond to different mannose content during Golgi glycosylation extensions of the RBD, as all bands merged to a single one of about 26 kDa after deglycosylation with endoglycosidase H, as previously reported by our group (Figure 3A). In addition, Figure 3B describes the increase in the total proteins and RBD concentration, as well as the percentage of the RBD in the total proteins every 12 h of the methanol induction phase corresponding to the fermentation performed with Strategy 2. Table 3 describes the evolution of the production parameters during the fermentation carried out with Strategy 2 in the 7 L stirred-tank bioreactor. It should be noted that the RBD concentration increased more than 20-fold during the induction, and the percentage of the RBD in the total proteins increased from 26.5% to 37.7% at the end of the induction phase. Under both strategies, a low RBD concentration was detected at the beginning of the induction (0 h). These amounts of recombinant protein were produced due to the RBD expression during the glycerol-fed-batch stage when this substrate was provided in a limited amount and during the adaptation phase when methanol was added. It is well known that both conditions allow the derepression of the AOX1 gene and, thus, the production of low levels of the recombinant protein [64–66].

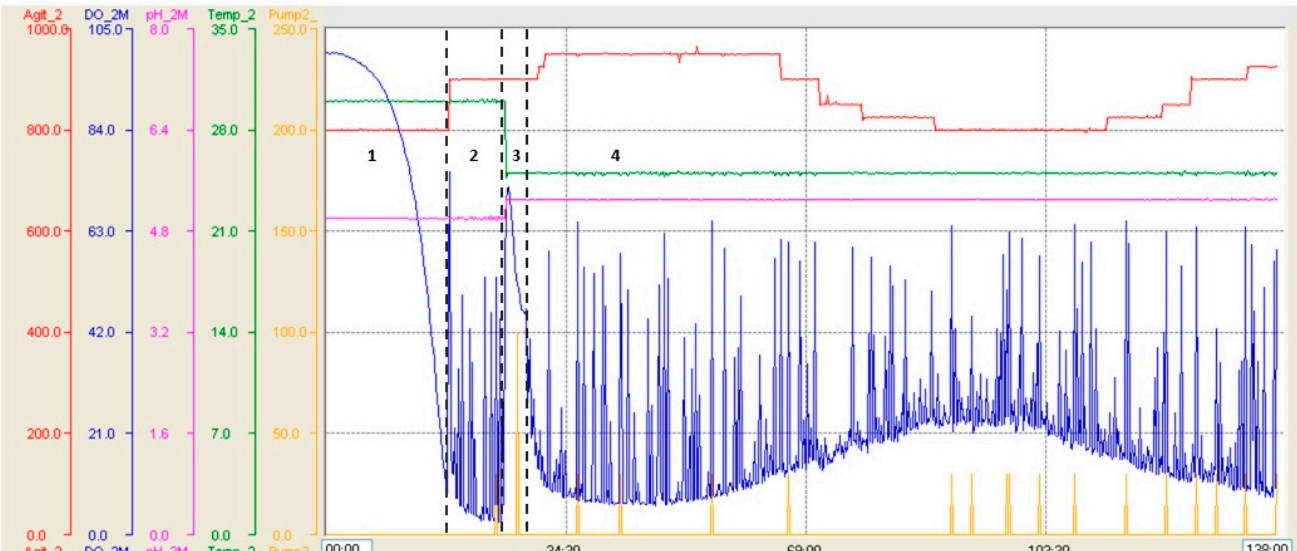

**Figure 2. Parameter profile of *P. pastoris* fermentation carried out with Strategy 2 in 7-L stirred-tank bioreactor.** Blue line: dissolved oxygen level (saturation percentage), green line: temperature (°C), pink line: pH, red line: stirring (RPM), yellow line: feeding (pumping percentage). 1: Batch phase, 2: glycerol-fed-batch phase, 3: transition phase, 4: induction phase.

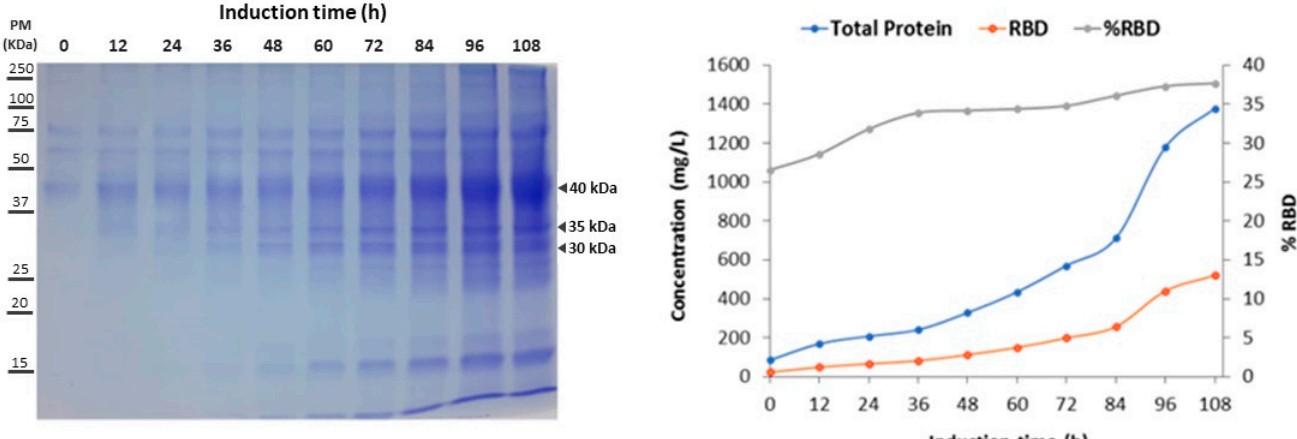

**Figure 3. SDS-PAGE analysis and production parameters of samples from the induction phase of 7-L bioreactor fermentation**. (**A**) SDS-PAGE protein profile of fermentation supernatants corresponding to methanol induction phase. Arrows indicate the molecular weights of the different glycosylated RBD variants. PM: Protein marker. (**B**) Variation in total protein concentration, RBD concentration, and RBD percentage in total protein during methanol induction.

**Table 3. Evolution of production parameters during fermentation carried out with Strategy 2 in 7-L stirred-tank bioreactor:** Detailed parameters every 12 h of methanol induction: total protein concentration (mg/L); RBD concentration (mg/L); RBD percentage with respect to total protein (%): RBD concentration increase (fold); $Y_{RBD/X}$: RBD yield based on biomass formation (mg/g); $_vP$: volumetric RBD productivity (mg RBD/L h).

| Induction Time (h) | Total Protein Concentration (mg/L) | RBD Concentration (mg/L) | RBD Percentage (%) | RBD Increase (fold) | Y RBD/x (mg/g) vP | vP (mg/L h) |
|---|---|---|---|---|---|---|
| 0 | 85.0 | 22.6 | 26.5 | 1.0 | 0.5 | 0.8 |
| 12 | 168.7 | 48.2 | 28.6 | 2.1 | 0.8 | 1.2 |
| 24 | 207.7 | 66.0 | 31.8 | 2.9 | 1.0 | 1.2 |
| 36 | 241.0 | 81.7 | 33.9 | 3.6 | 1.1 | 1.3 |
| 48 | 328.5 | 112.3 | 34.2 | 5.0 | 1.4 | 1.5 |
| 60 | 435.0 | 149.6 | 34.4 | 6.6 | 1.9 | 1.7 |
| 72 | 570.6 | 198.6 | 34.8 | 8.8 | 2.4 | 2.0 |
| 84 | 711.4 | 256.8 | 36.1 | 11.4 | 2.9 | 2.3 |
| 96 | 1178.3 | 439.8 | 37.3 | 19.5 | 4.9 | 3.5 |
| 108 | 1378.5 | 519.6 | 37.7 | 23.0 | 5.8 | 3.8 |

After fermentation in a 7 L bioreactor, RBD production was scaled up to a 14 L stirred-tank bioreactor with an initial volume of 4 L and using Strategy 2 as a four-step procedure. In the batch phase, a biomass concentration of 17.6 g DCW/L was achieved after 18 h, reaching similar $\mu_{max}$ and $Y_{X/S}$ values to those previously obtained. Next, in the fed-batch phase, glycerol feeding was performed for 8 h, in response to %DO with a saturation cut-off of 60%, obtaining a biomass concentration of 41.7 g DCW/L. The transition phase was started by supplying the culture with a dose of 4 g/L methanol as the sole carbon and energy source. A biomass level of 42.5 g DCW/L was achieved after 4 h. During the subsequent fed-batch induction phase, feeding of pure methanol was performed using a DO-stat strategy, in which methanol feeding was controlled in response to %DO with a saturation cut-off of 60%. After 115 h of methanol induction and a total fermentation process of 145 h, the fermented culture reached a final biomass level of 90.3 g DCW/L. At

this time, the total protein concentration in the fermented culture was 1032.7 mg/L, while the RBD concentration reached 533.4 mg/L, corresponding to 51.7% of the total proteins in the supernatant.

The total recombinant RBD was 2987 mg, as the volume of the supernatant was 5.6 L (Table 4). The RBD yield based on biomass ($Y_{RBD/X}$) at the end of fermentation was 5.9 mg RBD/g DCW; the volumetric RBD productivity ($_vP$) of the whole process reached 3.7 mg RBD/L h; the total RBD productivity ($_TP$) was 20.7 mg RBD/h (Table 4). Moreover, the specific RBD productivity ($_sP$) of the whole process reached 40.7 µg RBD/g DCW h, and the RBD yield based on methanol consumed ($Y_{RBD/S}$) was 1.9 mg RBD/g methanol at the end of the induction period. It is important to note that the values obtained for the RBD yield based on the biomass ($Y_{RBD/X}$), volumetric productivity ($_vP$), and specific productivity ($_sP$) of the whole process were similar to those obtained with the 7 L bioreactor applying Strategy 2, indicating that the scale-up was successful.

**Table 4. Evolution of production parameters during fermentation carried out with Strategy 2 in 14 L stirred-tank bioreactor:** Detailed parameters every 24 h of methanol induction: total protein concentration (mg/L); RBD concentration (mg/L); RBD percentage with respect to total protein (%); RBD concentration increase (fold); $Y_{RBD/X}$: RBD yield based on biomass formation (mg/g); $_vP$: volumetric RBD productivity (mg RBD/L h).

| Induction Time (h) | Total Protein Concentration (mg/L) | RBD Concentration (mg/L) | RBD Percentage (%) | RBD Increase (fold) | Y RBD/x (mg/g) vP | vP (mg/L h) |
|---|---|---|---|---|---|---|
| 0 | 120.3 | 37.3 | 31.0 | 1.0 | 1.0 | 1.2 |
| 24 | 169.7 | 65.2 | 38.4 | 1.7 | 1.1 | 1.2 |
| 48 | 237.0 | 101.2 | 42.7 | 2.7 | 1.4 | 1.3 |
| 72 | 415.3 | 188.1 | 45.3 | 5.0 | 2.4 | 1.8 |
| 96 | 669.2 | 321.8 | 48.1 | 8.6 | 3.9 | 2.6 |
| 115 | 1032.7 | 533.4 | 51.7 | 14.3 | 5.9 | 3.7 |

The parameter profile of the fermentation shown in Figure 4 specifically displays the variation of stirring, dissolved oxygen, and feeding during the process. It shows the decrease of the DO level in the batch phase until its abrupt increase (DO spike) and, later, the variations of dissolved oxygen that controlled the feeding of glycerol and methanol. In the DO-stat cultivation strategy, the input of the substrate was regulated by %DO. This means that, during the whole procedure, pulses of glycerol or methanol will be associated with fluctuations in %DO, which do not affect cell growth. The SDS-PAGE analysis revealed the increase in total protein and recombinant RBD during the induction phase with methanol (Figure 5A). It is worth mentioning that the glycosylated variants of the recombinant RBD corresponded to the diffuse bands of around 40, 35, and 30 kDa. Figure 5B shows the increase in total proteins and RBD concentration, as well as the percentage of RBD in total proteins every 24 h of the induction phase. The RBD concentration increased more than 14-fold, and the percentage of RBD in total proteins increased from 31% to 52% at the end of this phase.

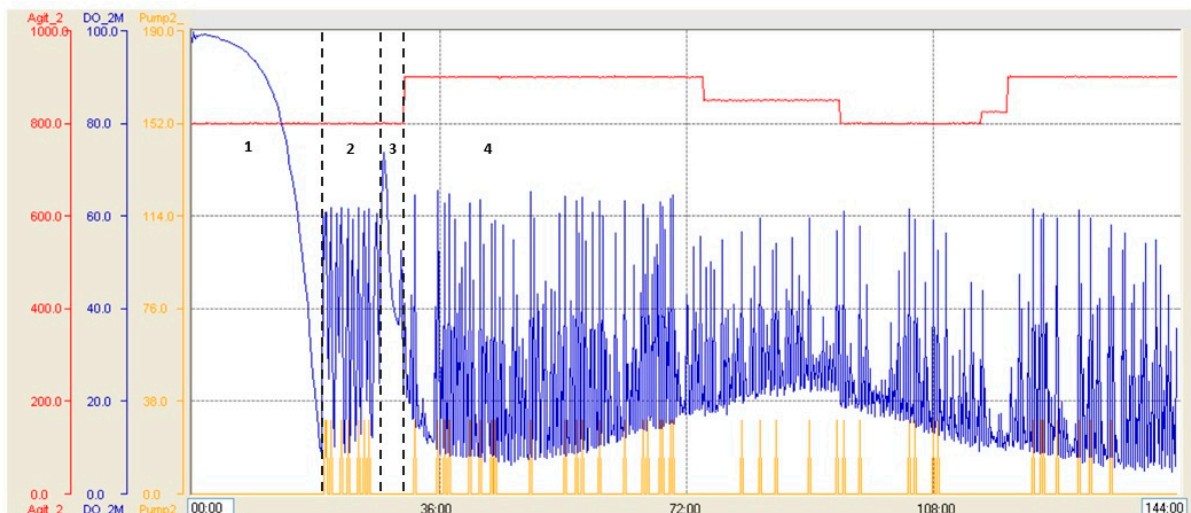

**Figure 4. Parameter profile of fermentation carried out in 14 L stirred-tank bioreactor with Strategy 2**. Blue line: dissolved oxygen level (saturation percentage), red line: stirring (RPM), yellow line: feeding (pumping percentage). 1: Batch phase, 2: glycerol-fed-batch phase, 3: transition phase, 4: induction phase.

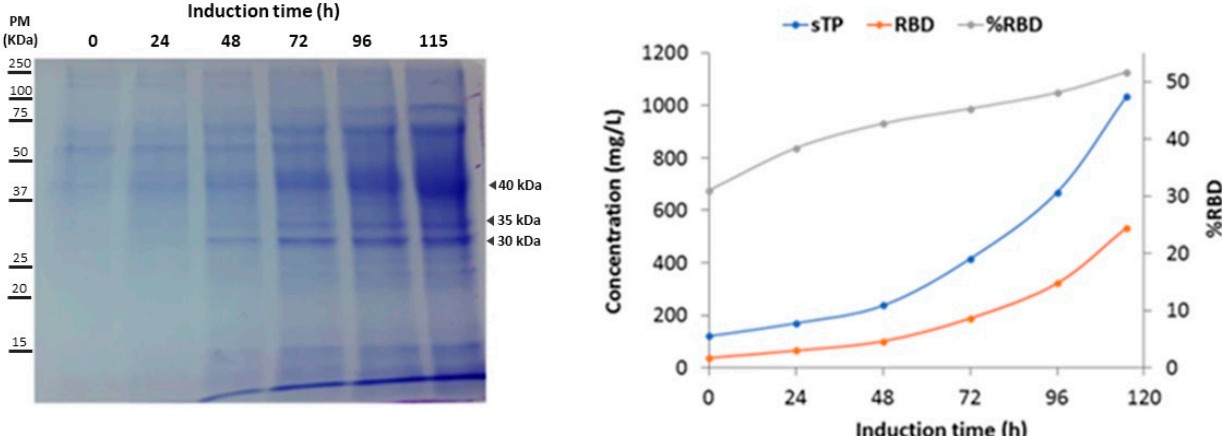

**Figure 5.** SDS-PAGE analysis and production parameters of the induction phase in a 14 L bioreactor fermentation performed with Strategy 2. (**A**) SDS-PAGE total protein profile of fermentation supernatants corresponding to induction phase. Arrows indicate the molecular weights of the different glycosylated RBD variants. PM: Protein marker. (**B**) Variation in total protein concentration, RBD concentration, and RBD percentage in total protein during methanol induction.

### 3.3. Purification and Analysis of RBD

The RBD was recovered from the fermentation supernatant as already described [13]. Briefly, 1–2 L fractions of the supernatant were purified in a single step using 20 mL of a $Ni^{2+}$-NTA affinity column. Pure RBD was eluted using imidazole as a competitor, and the protein was dialyzed with Tris 20 mM and NaCl 150 mM, pH 7.4. The protein analysis resulted in being identical to the one already characterized for the production at a smaller-scale fermentation, as the UV spectrum, mass spectrum, SDS-PAGE, and circular dichroism were indistinguishable from those previously reported by our group [13,14]. The protein batches of 7 L reported here were also obtained at more than 95% purity as determined by the HPLC analysis (Figure 6).

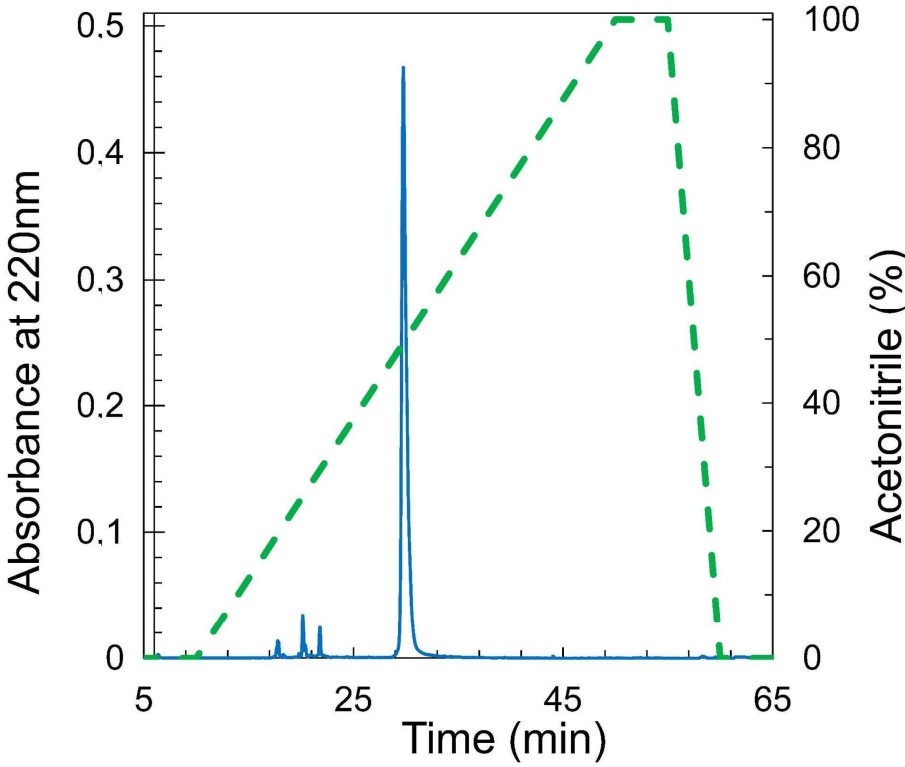

**Figure 6. Analysis by RP-HPLC of RBD produced in *P. pastoris* and purified by NTA-Ni²⁺.** Analytical C18 reverse-phase HPLC chromatogram for RBD produced in *P. pastoris* (20 μg) was obtained upon an ACN gradient 0 to 100% over 40 min (10–50 min of the run, dashed line) and with a mobile phase of 0.05% TFA. The integration of the main peak was 95% (filled line).

## 4. Discussion

As a quick response to the SARS-CoV-2 pandemic, several research groups around the world started to develop biological tools to provide raw materials for the diagnosis, treatment, and prevention of the disease. In such a scenario, during 2020, we reported the production of SARS-CoV-2 RBD using *P. pastoris* as a cell factory applying a preliminary, non-optimized bioreactor culture procedure. We also compared its structural features with RBD produced in mammalian HEK293T cells to verify that they were similar [13]. As the next step, in this article, we describe a rational procedure for the improvement of SARS-CoV2 RBD production using *P. pastoris* in 7 and 14 L bioreactors. To do that, two different culture strategies were tested, Strategy 2, a combination of batch and fed-batch using a DO-stat feeding, showing the best performance by allowing the obtention of more than 500 mg/L of raw yRBD in the culture broth. This cultivation procedure represents a simple, robust, scalable, and low-cost method since it involves the use of a stirred-tank bioreactor (STBR), a defined basal salt medium with simple carbon sources (glycerol and methanol), and the provision of oxygen exclusively from compressed air, avoiding the use of pure $O_2$ and its associated risks [67,68]. This affirmation is especially valid when comparing RBD production with other expression hosts such as eukaryotic cells [69,70]. Additionally, the use of a defined saline medium, avoiding or minimizing the requirement of complex undefined ingredients, allows the monitoring of component concentrations throughout the cultivation period, which is considered as a valuable feature for industrial processes [71]. All these advantages turn RBB produced in *P. pastoris* following the procedure reported here into an attractive molecule to be used in diagnostic tools or for vaccine developments, especially considering its cost and easy production.

Based on the specific growth rate (μ) obtained during the first 24 h of induction (0.072 h⁻¹) of the Erlenmeyer flask cultivation, it can be surmised that this recombinant clone behaves as a Mut⁺ strain, as expected. In this sense, Orman et al. [72] reported

a maximum specific growth rate ($\mu_{max}$) of 0.16 h$^{-1}$ for a recombinant *P. pastoris* clone expressing hGH in a defined medium with methanol as the sole carbon and energy source. Pla et al. [73], working with Mut$^+$ and Mut$^s$ clones expressing scFV, obtained a $\mu_{max}$ of 0.044 h$^{-1}$ and 0.015 h$^{-1}$, respectively. A recent study proposed that the use of Mut$^+$ phenotype is convenient for high levels of heterologous protein production considering that pAOX1 is induced not only by methanol, but also by its metabolites: formaldehyde and formate [74]. Therefore, a high methanol utilization results in a stronger pAOX1 induction, increasing heterologous protein production compared to a lower methanol utilization metabolism. These authors reported that the Mut$^+$ clone showed a specific β-galactosidase (heterologous protein) expression rate 5- and 10-fold higher than the Mut$^s$ and Mut$^-$ ones. In our work, the observed Mut$^+$ phenotype could be one of the reasons supporting the high level of RBD expression.

It is well established that an adequate induction phase design is crucial for high heterologous protein titers in *P. pastoris* when the AOX1 promoter is used [75,76]. In our procedure, the transition phase from glycerol was performed as a pure methanol pulse to achieve a concentration of 4 g/L in the culture broth followed by the induction phase under a DO-stat feeding for 108 h. The criteria underlying this design are related to providing enough methanol for adaptation and induction, minimizing the risk of methanol accumulation, as well as keeping $O_2$ demand and heat production controlled. Some methanol feeding strategies keeping its concentration constant or within a defined range have been reported for heterologous protein production in *P. pastoris* and *P. methanolica* [77,78]. These strategies allow a strong and constant AOX1 induction, while avoiding methanol accumulation and the consequent cell intoxication. Several analytical methods to achieve this feeding profile have been developed [79,80]. Most of them require specific and expensive equipment. Considering such arguments, the strategy applied in this work emerges as an alternative procedure when permanent methanol concentration surveillance is not available. The induction strategy used in this work is based on and shares some features with those reported by Yamawaki et al. [44] for the production of an antibody fragment (scFv). For induction, these authors combined a stage in which the methanol concentration was kept at 15.7 g/L for 5 h (controlled by a methanol sensor feedback) followed by a DO-stat for 36 h (total induction time: 41 h). Under these conditions, 247 mg/L of recombinant protein was obtained. In the case of the process reported here, cells in the bioreactor were exposed to a significant, but non-toxic methanol level (4 g/L) adequate for AOX1 activation. The adaptation stage end was deducted from a $O_2$ spike, and indicator of methanol exhaustion and a signal to start the methanol-fed-batch under a DO-stat strategy. Therefore, we combined an adaptation stage with a methanol pulse and the subsequent induction of the RBD expression with methanol feeding in response to %DO in the culture broth as a production strategy.

Using this procedure, around 500 mg/L of RBD was obtained, solubilized in the culture broth. Further purification steps resulted in the obtention of 206.4 mg/L of pure protein (95%). This amount of recombinant protein was significantly higher than those obtained and reported previously (96.1 mg/L of total RBD and 45 mg/L of 95% pure protein), representing an ~5-fold improvement [13].

Other authors [57] reported the production of 400 mg/L of a recombinant SARS-CoV RBD219-N1 using *P. pastoris* X-33. For this purpose, these authors developed a multistage process including several feed flows and gradients. In that case, the induction stage took ~70 h. During the process, %DO was maintained above 30%, adjusting the gas provision and agitation. These authors did not report the gas provision source (air or $O_2$).

In a recent article [81], the production of a modified SARS-CoV-2 RBD using *P. pastoris* X-33 at a 50 L scale was reported. The production process was based on a saline medium containing yeast extract and other elements (histidine, biotin, myo-innositol, calcium pantothenate, pyridoxal hydrochloride, thiamine di-hydrochloride, and nicotinic acid). The process involved the use of constant feedings rates and yielded a dry cell weight of 58.15 g/L and 68.38 mg/L of the RBD. Downstream consisted of an IMAC (Cu$^{2+}$) followed

by a semi-preparative RP-HPLC. After that, the recombinant RBD showed a purity equal to or higher than 98%. The bioreactor culture lasted for 38 to 48 h. Both the DCW and RBD concentration were lower than those obtained under the proposed procedure (90 g/L vs. 58 g/L and 500 mg/L vs. 68 mg/L). However, the fermentation time was shorter in the procedure reported by Limonta-Fernandez et al. (2022) [81], providing a potential advantage for industrial production.

Methanol metabolism by alcohol oxidase is a process that requires high levels of oxygen and releases a large amount of energy as heat. *P. pastoris* high-density cultures using this substrate are usually carried out by providing pure oxygen. The DO-stat strategy applied for both the glycerol-fed-batch and methanol induction resulted in a moderate $O_2$ consumption and a gradual heat production.

As was stated by Cardoso et al. (2020) [82], the costs associated with heterologous protein production using microorganisms are driven by medium composition and cooling. The cultivation method proposed in this work is coherent with this affirmation, considering that it involves the use of a simple and relatively inexpensive medium and a DO stat strategy, thus diminished oxygenation and cooling requirements.

In the case of prokaryotic hosts, the RBD is usually obtained as a non-glycosylated, non-folded protein. For this reason, re-folding is needed during downstream processing. Related to that, the production of the RBD was reported using *E. coli* as an expression system. He et al. (2021) [83] developed a production method using a BL21 strain obtaining the most the RDB in inclusion bodies. After solubilization and renaturalization, they recovered 65.2% of the produced RBD by Nickel affinity chromatography, reporting a production yield of 13.3 mg/L. Meena et al. [84] also informed RBD production in bacterial cells as inclusion bodies. These authors cultivated an *E. coli* strain carrying the RBD gene in n Erlenmeyer flask containing 1 L of a complex undefined media (a modification of the Luria Bertani broth), ampicillin, and IPTG induction. After cultivation, they obtained 62.10 mg of raw RBD as inclusion bodies from 1 L of culture. After that, several downstream steps were needed, including solubilization at pH 12.5 and 3, refolding, and DEAE chromatography.

The RBD obtained from this process is a glycosylated protein showing three main variants (30, 35, and 40 kDa). These variants present the same primary structure differing only in the glycosylation pattern [13]. Beyond its crucial function in protein folding, glycosylation represents an issue of special concern for RBD industrial production, since the intra- and inter-batch heterogeneity could be considered as a drawback for standardization and regulation compliance [85]. Even though the glycosylation status was heterogeneous, the protein behaves identically to the RBD expressed in mammalian HEK-293T cells, in its conformation, stability, and immunogenic properties. Moreover, antibodies raised when using our *P. pastoris* produced the RBD cross-reacted with the RBD produced in mammalian cells and vice versa [14]. Further research is needed to clarify the impact of this post-translational modification on biological activity and to improve protein homogeneity. We are currently working on several factors to improve protein homogeneity. Regarding the addition of a Hisx6 tag, it is worth mentioning that, even though the purification methods of a non-tagged RBD were reported [57,86], our strategy is based on the addition of a tag that allows purification to a high purity in a single step with the capability to be easily removed, prioritizing procedure simplicity and robustness. Our construction bears a Sortase A sequence at the N-terminal of the Hisx6 tag, which allows its removal in a transpeptidation reaction to a multimer [13] or Gly3 trisaccharide or even water (manuscript in preparation). The strategy provides the versatility to—depending on the use of the protein: diagnosis, generation of neutralizing antibodies, or vaccine development—remove or leave the tag in a single step.

In summary, in this work, we presented an efficient and reliable scaling up method for the production of a low-cost RBD antigen using the methylotrophic yeast *P. pastoris*, with multiple advantages. One possible drawback of this procedure is the extension of the induction stage. This extension was partially caused by the DO-stat strategy, which involves a kind of "dead time" among methanol feeding pulses. These periods are minutes in which

no or very low levels of methanol are available in the culture broth, probably resulting in a waste of time for the metabolic machinery of *P. pastoris*. As an alternative, a strategy in which the methanol concentration is kept constant could be more effective. However, this strategy would require a method for real-time methanol concentration evaluation, through a GC-associated method or a specific detector, which requires specific equipment. As Strategy 2 resulted in the obtention of a well-folded, immunologically active RBD, we propose the method provided here to scale up the production of RBD.

**5. Conclusions**

Industrial production of recombinant proteins, especially those for pharmaceutical products, draw upon different living organisms as hosts, harnessing their advantages while dealing with their disadvantages [87,88]. There is no universal host for this task, and each protein can be produced "conveniently" in a process meeting a swarm of factors, the protein features (structure and post-translational modifications), production performance and costs, protein application, and legal regulations being some of the most-relevant. Here, we proposed such a convenient process for RBD production.

**Author Contributions:** D.G.N.: methodology, investigation, writing-original draft, visualization. C.D.: conceptualization, investigation, methodology, resources, writing—review and editing, funding acquisition, project administration, J.S.: funding acquisition, writing—review and editing, project administration, T.I.-H.: methodology, investigation. F.P.: methodology, investigation. D.E.W.: methodology, resources. H.G.: methodology. A.D.N.: funding acquisition, conceptualization, project administration, writing—review and editing. E.R.: methodology and validation. C.P.: methodology. L.A.M.R.: conceptualization, investigation, methodology, writing—original draft, project administration. All authors have read and agreed to the published version of the manuscript.

**Funding:** This work was supported partially by the grants IP COVID-19–234 and PICTO 2021-0007 provided by the ANPCYT (Argentina).

**Institutional Review Board Statement:** Not applicable.

**Informed Consent Statement:** Not applicable.

**Data Availability Statement:** Data will be made available upon request.

**Acknowledgments:** We would like to thank the Consorcio Argentino AntiCovid for helpful discussions and collaborative work during the pandemic.

**Conflicts of Interest:** The authors declare that they have no known competing financial interest or personal relationships that could have appeared to influence the work reported in this paper. The strain and some steps of the processes presented are included in the pending Patents PCT/US22/43578 and AR P210102579.

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
