# Peer review of "Development of a Cost-Effective Process for the Heterologous Production of SARS-CoV-2 Spike Receptor Binding Domain Using Pichia pastoris in Stirred-Tank Bioreactor"

_fermentation, doi:10.3390/fermentation9060497_

Round 1

Reviewer 1 Report

In this manuscript, authors developed an upstream process to produce a his-tagged receptor binding domain (RBD) of SARS-CoV-2 spike protein using P. pastoris as the expression host. The optimized process was able to produce ~500 mg RBD/L of culture broth in the bioreactor. Overall, the manuscript is relevant to the current pandemic, however, several comments have been listed below for the authors to address:

Line 49. Could authors clarify “semester” in the sentence? The second half of 2022?

Line 150. The authors chose residues 319-537 of SARS-CoV2 S protein for expression in P. pastoris. Could authors specify the variant of the SARS-CoV2 the residues were chosen from? Additionally, could the authors explain why the sequence was chosen? Many publications suggest that the RBD of SARS-CoV-2 seems to be slightly further downstream to the sequence authors proposed, e.g., Residues 333-527 in Lan et al, Nature 2020 (https://pubmed.ncbi.nlm.nih.gov/32225176/), residues 331-528 in Premkumar et al, Sci Immunol. 2020 (https://pubmed.ncbi.nlm.nih.gov/32527802/), residues 332-549 in Lee et al., Appl Microbiol Biotechnol. 2021 (https://pubmed.ncbi.nlm.nih.gov/33959781/) and residues 332-533 in Chen et al., Protein Expr Purif.  2022 (https://pubmed.ncbi.nlm.nih.gov/34688919/)

Lines 170-172. Authors used densitometry to evaluate the RBD content in culture broth and purified fractions. Did authors use any protein standard for such quantification?

Line 257. Authors stated that UV spectrophotometry was used to quantify protein content. Could authors specify the wavelength selected for the measurement? Was it 280nm? Additionally, was the extinction coefficient incorporated for such measurement? Or did authors simply use the common assumption, i.e., 1OD ~ 1mg/mL?

Line 258. Please correct “spectrometry” to “spectrophotometry”.

Table 1. Authors detected 1.6 mg/L of RBD protein even when the induction was not initiated (at 0 hours). Similarly in Tables 3 and 4, at 0-hour induction time, 22 and 37.3 mg/L of RBD were detected, respectively. Shouldn’t protein only be expressed after induction? Please justify.

Figures 2 and 4. The DO output for both bioreactor runs fluctuated severely. Would it impact the cell culture?

Figure 3A. How do we know that the 30-, 35- and 40- kDa bands observed on the SDS-PAGE are not host cell protein bands? Did authors perform an anti-his-tag or anti-RBD western blot to confirm? If not, please consider performing such an experiment.

There is no functionality data to support that the expressed RBD maintains its structure or function. Please consider performing additional experiments, such as ACE-2 binding study using BLI, SPR or ELISA to confirm the RBD functionality.

Line 529. In the discussion section, the authors referenced two articles (refs 79 and 80). These articles talked about RBD of SARS-CoV not SARS-CoV-2. Please update accordingly.

On the same note, Chen et al. (Protein Expr Purif.  2022; https://pubmed.ncbi.nlm.nih.gov/34688919/) reported that RBD could be expressed at the level of 490 mg/L of culture broth for only 3 days of induction. The operation time was shorter yet the production yield was similar. Similarly, Lee et al. (Appl Microbiol Biotechnol. 2021, https://pubmed.ncbi.nlm.nih.gov/33959781/) also reported that with the addition of glycerol fed-batch, the raw yield of RBD could reach 533 mg/L of fermentation supernatant in a 1L bioreactor for 70-hour induction. These proteins seem to be more homogenous or at least did not seem to show severe hyperglycosylation as what authors have pointed out. That being said, could the authors please justify why their production process is superior to the earlier publications?

Lines 573-574. Due to hyperglycosylation, a commonly seen phenomenon in yeast-expressed heterologous protein, authors indicated that it could cause drawbacks for standardization and regulations compliance. Thus, further ongoing improvement was conducted toward protein homogeneity. Could authors briefly discuss the improvement plan? e.g., change different Pichia strains?

Author Response

This paper describes methodology to produce the Receptor Binding Protein (RBD) of COVID-19 in P. pastoris, which would be beneficial to fight against the virus for various purposes. This study is likely suitable for the aim of the journal Fermentation. However, revisions are required for publication. My comments are followings:

1) Quality of figures 2 and 4 are poor. These may be screenshot images, so numbers and words are not clear.

We thank the reviewer for the comment. Figures 2 and 4 have been replaced for better quality ones.

2) In SDS-PAGE images in Figure 3 and 5, molecular weight markers are required.

Now positions of the MWM have been added to the SDS-PAGES

3) Figure 6: I think the HPLC result shown in this figure is not sufficient to show that the authors really purified the protein. I understand this experiment was performed and shown before. But control sample for HPCL and SDS-PAGE or MS results are required for this figure.

We thank the reviewer for the comment. The purpose of this manuscript was not to make a full characterization of the protein produced, as such analysis has already been published in Sci. Rep. 2020. Rather, the aim of this work was to show the feasibility to scale the production of the protein to  better yields. To make this clearer for the readers and avoid repeating the published results now it states in the text (Results,  section 3.4, Purification and analysis of RBD) (lines 479 to 486): 

…..RBD was recovered from the fermentation supernatant as already described [14]. Briefly, 1-2 liter fractions of supernatant were purified in a single step using 20 mL of a Ni2+-NTA affinity column. Pure RBD was eluted using imidazol as a competitor, and the protein was dialyzed with Tris 20 mM, NaCl 150 mM, pH 7.4. The protein analysis resulted identical to the one already characterized for the production at a smaller scale fermentation, as UV-spectrum, mass spectrum, SDS-PAGE and circular dichroism were indistinguishable from those previously reported by our group [13, 14]. Protein from batches of 7L reported here were also obtained at more than 95% purity as determined by HPLC analysis …

4) Recent paper describes production of RBD without tagging in P. pastoris (doi.org/10.1038/s41598-023-32021-9) and I wondered if addition of his tag influences further applicational procedures. This information would be worth being described in this paper.

We thank the reviewer for the opportunity to clarify this point. A new sentence and the citation to the mentioned article have been added. (Discussion, lines 608-618)

It now states in the discussion: …. “We are currently working on several factors to improve protein homogeneity. Regarding the addition of a His6 x tag, it is worth mentioning that even though purification methods of a non-tagged RBD were reported [58,86], our strategy is based on the addition of a tag that allows purification to a high purity in a single step with the capability to be easily removed, prioritizing procedure simplicity and robustness.  Our construction bears a Sortase A sequence at the N-terminal of the Hisx6 tag that allows its removal in a transpeptidation reaction to a multimer [13] or Gly3 trisaccharide or even water (manuscript in preparation). The strategy provides the versatility to –depending on the use of the protein: diagnosis, generation of neutralizing antibodies or a vaccine development- remove or leave the tag in a single step.   ”... 

Minor point:

-Please check the manuscript carefully since I found several typo mistakes. 

The manuscript has been carefully revised and (hopefully) all the mistakes have been corrected.

For example:

Line 68, it’s was replaced for “its”.

Line 252-259, Different font is used probably due to copy/paste. 

Font style was harmonized.

Line 315, 1,4-fold was replaced for 1.4-fold

Line 486, Figure 6 (Font size is larger than the others)

Font size was harmonized 

h (hour) and hs (hours) are not appropriately used.

Hs was replaced for h (line 207 x2, 212, 546 x2, 547)

Reviewer 2 Report

This paper describes methodology to produce the Receptor Binding Protein (RBD) of COVID-19 in P. pastoris, which would be beneficial to fight against the virus for various purposes. This study is likely suitable for the aim of the journal Fermentation. However, revisions are required for publication. My comments are followings:

1) Quality of figures 2 and 4 are poor. These may be screenshot images, so numbers and words are not clear.

2) In SDS-PAGE images in Figure 3 and5, molecular weight markers are required.

3) Figure 6: I think the HPLC result shown in this figure is not sufficient to show that the authors really purified the protein. I understand this experiment was performed and shown before. But control sample for HPCL and SDS-PAGE or MS results are required for this figure.

4) Recent paper describes production of RBD without tagging in P. pastoris (doi.org/10.1038/s41598-023-32021-9) and I wondered if addition of his tag influences further applicational procedures. This information would be worth being described in this paper.

Minor point:

Please check the manuscript carefully since I found several typo mistakes. 

For example:

Line 68, it’s

Line 251-258, Different font is used probably due to copy/paste.

Line 301, 1,4-fold

Line 459, Figure 6 (Font size is larger than the others)

h (hour) and hs (hours) are not appropriately used.

Author Response

The work entitled “Development of a cost-effective process for the heterologous production of SARS-CoV-2 spike receptor binding domain (RBD) using Pichia pastoris in stirred-tank bioreactor”, authors are Diego Noseda, Cecilia D’Alessio, Javier Santos, Tommy Idrovo-Hidalgo, Florencia Pignataro, Diana E Wetzler, Hernán Gentilli, Alejandro D Nadra, Ernesto Roman, Carlos Paván and Lucas AM Ruberto,is devoted to biotechnological production of the SARS-CoV-2 spike protein for diagnosis, treatment and prevention of Covid-19. The topic of the paper is practically useful and important. The producers of the recombinant target protein were yeasts Pichia pastoris. The selected P. pastoris X-33 strain was completely biologically safe, fast growing, contained the strong alcohol oxidase 1 (AOX1) promoter and was able to modify RBD after translation (glycosylation, proteolytic processing and disulfide bonds formation). The RBD production was measured at three levels: in flasks with 50 ml of the cultivation medium and a stirred tank-bioreactor with 7 and 14 L vessels. The most efficient regime of feeding and induction was developed, which resulted in production of 500 mg/L of raw yRBD. Figures and Tables are very informative and provide with all necessary details. Relevant comparisons of the obtained results with the published literature were performed, and cost-effectiveness of the suggested scaling-up procedure was shown.

Major revision

Although adequate comparisons of the obtained results with results of previous works and other literature are conducted, what are really significant novel findings in this work in comparison with the reference [14], section “Production of RBD by fermentation in bioreactor” (Argentinian AntiCovid Consortium Structural and Functional Comparison of SARS-CoV-2-Spike Receptor Binding Domain 643 Produced in Pichia Pastoris and Mammalian Cells. Sci. Rep. 2020, 10, 21779, doi:10.1038/s41598-020-78711-6)? In the Sci. Rep. work, the same 4-stage procedure with glycerol-methanol feeding and control of O2 demand was used for recombinant RPD production by P. pastoris in the same 7-L stirred-tank bioreactor was used.

We thank the reviewer for the opportunity of clarifying this point. The purpose of the manuscript in Sci. Rep 2020 was to compare RBD expressed in Pichia pastoris with that produced in a mammalian system, and not to optimize the production of the protein. Although the rationale of a production developed at that moment included essentially the same four stages (as is clarified in section 2.7, line 213-215), the procedure described here is different and includes modifications that allowed a much higher RBD concentration (519,6 mg/L) at an even higher scale with lower costs. For example, the new procedure includes a transition stage using only methanol (4 g/L) while the previously reported method used a mixture (3;1) glycerol:methanol. In the new manuscript, we reported the performance of the adjusted procedure using 7 and 14 L  bioreactors, while in the previous report (2020) we only applied the preliminar procedure at a 7 L scale. To make it clearer for the readers the first paragraph of the discussion (lines 493-500)  was modified to:

“As a quick response to the SARS-CoV-2 pandemic, several research groups around the world started to develop biological tools to provide raw materials for diagnosis, treatment, and prevention of the disease. In such a scenario during 2020 we reported the production of SARS-CoV-2 RBD domain using P. pastoris as a cell factory applying a preliminary, non-optimized bioreactor culture procedure. We also compared its structural features with RBD produced in mammalian HEK293T cells to verify that they were similar [14]. As the next step, in this article, we describe a rational procedure for the improvement of SARS-CoV2 RBD production using P. pastoris in 7 and 14 L bioreactors.”

Minor revisions

-p. 2, line 87 – This is the first appearance of the abbreviation DWC. It should be described.

We apologize for the omission. The description of the abbreviation has now been  added and updated 

-p. 4, line 185 – What concentration of sulfuric acid was used?

We apologize for the omission. A concentration of 98% w/w was used. This information has been added to Materials and Methods

-p. 5-6, lines 252-259 – Please, check font style.

Font style was homogenized.

-p. 6, line 259 – It is recommended to add details of the HPLC analysis. HPLC conditions are described in the Figure 6 capture but it is recommended to transfer these details to Materials and Methods.

The following information regarding the HPLC procedure was added (lines 263-267)

The purity of recombinant RBD was analyzed by reverse-phase HPLC using an analytical C18 column (Higgins Analytical, Inc. U.S.A.) and a JASCO system (equipped with an autoinjector, an oven at 25 °C and a UV detector). Elution was made at a 1.0 mL/min flow using a ACN gradient from 0 to 100% over 40 min (10- 50 min of the run) with a mobile phase of 0.05% TFA….

-What statistical methods for the data processing were used?

A paragraph describing statistical analysis was added to the manuscript (line 269-273)

Data for cell density, total protein concentration and RBD concentration were taken by triplicate. Media and Standard deviations were calculated.  Statistical significance was evaluated by the Student’s t-test using the Infostat V2020 Software. Differences were considered significant if p < 0.05….

-p. 6, Figure 1 – What are P1 and P2? They are not indicated on the figure. In B, Induccion should be corrected as Induction. Error bars are not visible. It is recommended to reduce the size of marks to show error bars, even if they are short and SD are low. In B, the decimal separator on the scale Specific growth rate should be point “.”.

Figure 1 was modified according to the reviewer's suggestions. Error bars were added and the decimal separator was changed.

-p. 7, Table 1 caption – Please, italisize P. pastoris.

“P. pastoris” was italicized in Table 1 caption.

-p. 9, Table 2 caption – Pichia pastoris should be corrected as P. pastoris. It is not the first appearance of the microorganism in the text.

Pichia pastoris was changed to P. pastoris in the Table 2 caption.

-Figures 2, 4 – The resolution of pictures should be increased, if possible.

Figures 2 and 4 were replaced with better-resolution ones.

-Sections 3.2 and 3.3 could be combined in one section for better comparison of strategies 1 and 2 and their effects in 7- and 14-L stirred-tank reactors.

The sections have been combined into a single one entitled “3.2. Production of recombinant RBD in 7-L and 14-L stirred bioreactors”

Reviewer 3 Report

The work entitled “Development of a cost-effective process for the heterologous production of SARS-CoV-2 spike receptor binding domain (RBD) using Pichia pastoris in stirred-tank bioreactor, authors are Diego Noseda, Cecilia D’Alessio, Javier Santos, Tommy Idrovo-Hidalgo, Florencia Pignataro, Diana E Wetzler, Hernán Gentilli, Alejandro D Nadra, Ernesto Roman, Carlos Paván and Lucas AM Ruberto,

is devoted to biotechnological production of the SARS-CoV-2 spike protein for diagnosis, treatment and prevention of Covid-19. The topic of the paper is practically useful and important. The producers of the recombinant target protein were yeasts Pichia pastoris. The selected P. pastoris X-33 strain was completely biologically safe, fast growing, contained the strong alcohol oxidase 1 (AOX1) promoter and was able to modify RBD after translation (glycosylation, proteolytic processing and disulfide bonds formation). The RBD production was measured at three levels: in flasks with 50 ml of the cultivation medium and a stirred tank-bioreactor with 7 and 14 L vessels. The most efficient regime of feeding and induction was developed, which resulted in production of 500 mg/L of raw yRBD. Figures and Tables are very informative and provide with all necessary details. Relevant comparisons of the obtained results with the published literature were performed, and cost-effectiveness of the suggested scaling-up procedure was shown.

Major revision

Although adequate comparisons of the obtained results with results of previous works and other literature are conducted, what are really significant novel findings in this work in comparison with the reference [14], section “Production of RBD by fermentation in bioreactor” (Argentinian AntiCovid Consortium Structural and Functional Comparison of SARS-CoV-2-Spike Receptor Binding Domain 643 Produced in Pichia Pastoris and Mammalian Cells. Sci. Rep. 2020, 10, 21779, doi:10.1038/s41598-020-78711-6)? In the Sci. Rep. work, the same 4-stage procedure with glycerol-methanol feeding and control of O2 demand was used for recombinant RPD production by P. pastoris in the same 7-L stirred-tank bioreactor was used.

Minor revisions

p. 2, line 87 – This is the first appearance of the abbreviation DWC. It should be described.

p. 4, line 184 – What concentration of sulfuric acid was used?

p. 5-6, lines 251-258 – Please, check font style.

p. 6, line 259 – It is recommended to add details of the HPLC analysis. HPLC conditions are described in the Figure 6 capture but it is recommended to transfer these details to Materials and Methods.

What statistical methods for the data processing were used?

p. 6, Figure 1 – What are P1 and P2? They are not indicated on the figure. In B, Induccion should be corrected as Induction. Error bars are not visible. It is recommended to reduce the size of marks to show error bars, even if they are short and SD are low. In B, the decimal separator on the scale Specific growth rate should be point “.”.

p. 7, Table 1 caption – Please, italisize P. pastoris.

p. 9, Table 2 caption – Pichia pastoris should be corrected as P. pastoris. It is not the first appearance of the microorganism in the text.

Figures 2, 4 – The resolution of pictures should be increased, if possible.

Sections 3.2 and 3.3 could be combined in one section for better comparison of strategies 1 and 2 and their effects in 7- and 14-L stirred-tank reactors.

Summary

Major revision.

Few spelling inaccuracies.

Author Response

In this manuscript, authors developed an upstream process to produce a his-tagged receptor binding domain (RBD) of SARS-CoV-2 spike protein using P. pastoris as the expression host. The optimized process was able to produce ~500 mg RBD/L of culture broth in the bioreactor. Overall, the manuscript is relevant to the current pandemic, however, several comments have been listed below for the authors to address:

  • Line 49. Could authors clarify “semester” in the sentence? The second half of 2022?

We apologize for the confusing sentence. In the revised version “Semester” was changed to “second half of 2022” (line 49).

-Line 150. The authors chose residues 319-537 of SARS-CoV2 S protein for expression in P. pastoris. Could authors specify the variant of the SARS-CoV2 the residues were chosen from? Additionally, could the authors explain why the sequence was chosen? Many publications suggest that the RBD of SARS-CoV-2 seems to be slightly further downstream to the sequence authors proposed, e.g., Residues 333-527 in Lan et al, Nature 2020 (https://pubmed.ncbi.nlm.nih.gov/32225176/), residues 331-528 in Premkumar et al, Sci Immunol. 2020 (https://pubmed.ncbi.nlm.nih.gov/32527802/), residues 332-549 in Lee et al., Appl Microbiol Biotechnol. 2021 (https://pubmed.ncbi.nlm.nih.gov/33959781/) and residues 332-533 in Chen et al., Protein Expr Purif.  2022 (https://pubmed.ncbi.nlm.nih.gov/34688919/)

We thank the reviewer for the comment. The aim of this work was to scale up the expression of yRBD already published at a smaller scale [14].  To avoid redundancy, we focused here on the bioreactor cultivation optimization.  The following sentence (lines 149-150) reflects this idea:

 “The RBD sequence and plasmid construction, as well as the P. pastoris strain used in this work, were the same described previously [13,14] “

-Lines 170-172. Authors used densitometry to evaluate the RBD content in culture broth and purified fractions. Did authors use any protein standard for such quantification?

Yes indeed, the quantification was performed by comparing the densitometry with known standards of BSA.

-Line 257. Authors stated that UV spectrophotometry was used to quantify protein content. Could authors specify the wavelength selected for the measurement? Was it 280nm? Additionally, was the extinction coefficient incorporated for such measurement? Or did authors simply use the common assumption, i.e., 1OD ~ 1mg/mL?

To clarify this point, the following information was now added to the text (lines 260-262).  

…” Absorption spectra (240-340 nm range, using a 0.1-nm sampling interval) were acquired at 20o C​ with a JASCO V730 BIO spectrophotometer (Japan). RBD concentration was determined using:​ ?​ = 33850 M​–1 cm​–1 (Abs ​ = 1.304 for a 1 mg mL​-1 protein solution).…“   

-Line 258. Please correct “spectrometry” to “spectrophotometry”.

The term “spectrometry” was removed and replaced by the phrase proposed in the previous point. 

-Table 1. Authors detected 1.6 mg/L of RBD protein even when the induction was not initiated (at 0 hours). Similarly in Tables 3 and 4, at 0-hour induction time, 22 and 37.3 mg/L of RBD were detected, respectively. Shouldn’t protein only be expressed after induction? Please justify.

At the end of the growth stage in Erlenmeyer cultures as well as during the glycerol fed-batch stage in bioreactor fermentations, the promoter AOX1 is partially derepressed due to limited glycerol availability, resulting in a low level of recombinant protein expression thus allowing the detection of RBD in low concentrations at the beginning of methanol induction phase (0 h). To clarify this important point to the readers it now states in the text (line 390-396):

….”Under both strategies, a low RBD concentration was detected at the beginning of the induction (0 h). This amount of recombinant protein is produced due to RBD expression during the glycerol fed-batch stage when this substrate is provided in a limited amount, and during the adaptation phase when methanol is the only carbon source. It is well-known that both conditions allow the derepression of the AOX1 gene and thus, the production of low levels of the recombinant protein [64–66]”....

-Figures 2 and 4. The DO output for both bioreactor runs fluctuated severely. Would it impact the cell culture?

The % DO level in bioreactor culture fluctuated in a range between 10 and 60% during fed-batch phases because the feeding of both glycerol and methanol was performed using a DO-stat strategy, in which the feeding of these substrates was in fact regulated with the % DO. In this cultivation strategy, the fluctuation in DO is absolutely normal  without any significant effect on cells (Konstantinov K, Kishimoto M, Seki T, Yoshida T. A balanced DO-stat and its application to the control of acetic acid excretion by recombinant Escherichia coli. Biotechnol Bioeng. 1990 Oct 5;36(7):750-8. doi: 10.1002/bit.260360714. PMID: 18597268). The cultures in the bioreactor achieved high levels of biomass using this fermentation strategy,indicating that DO fluctuation has no significant impact on cell growth. To make this point clearer for the readers it now states in the new version (lines 449-452):

…”In the DO-stat cultivation strategy, the input of substrate is regulated by %DO. This means that during the whole procedure, pulses of glycerol or methanol are associated with  fluctuations in DO, which do not affect cell growth”...

-Figure 3A. How do we know that the 30-, 35- and 40- kDa bands observed on the SDS-PAGE are not host cell protein bands? Did authors perform an anti-his-tag or anti-RBD western blot to confirm? If not, please consider performing such an experiment.

The identity of the 30- ,35-, and 40- kDa  bands has been confirmed by anti-his-tag western blot. The experiment showed that those bands correspond to glycosylated variants of recombinant RBD. Such confirmation was previously reported and published by our group (Argentinian AntiCovid Consortium. Structural and functional comparison of SARS-CoV-2-spike receptor binding domain produced in Pichia pastoris and mammalian cells. Sci Rep. 2020 Dec 11;10(1):21779. doi: 10.1038/s41598-020-78711-6. PMID: 33311634; PMCID: PMC7732851). To avoid any doubt to the readers, it now states in the text (lines 381-384): 

…”The three bands of 30- ,35-, and 40- kDa correspond to different mannose content during Golgi glycosylation extensions of RBD, as all bands merge to a single one of about 26 kDa after deglycosylation with endoglycosidase H, as previously reported by our group [14]”...

-There is no functionality data to support that the expressed RBD maintains its structure or function. Please consider performing additional experiments, such as ACE-2 binding study using BLI, SPR or ELISA to confirm the RBD functionality.

We thank the reviewer for the comment. The purpose of this manuscript was not to make a full characterization of the protein produced, as such analysis has already been published in Sci. Rep. 2020. Rather, the aim of this work was to show the feasibility to scale to better yields the production of the protein. To make this clearer for the readers and avoid repeating the published results now it states in the text (Results,  section 3.4, Purification and analysis of RBD)(lines 479 to 486): 

…..RBD was recovered from the fermentation supernatant as already described [14]. Briefly, 1-2 liter fractions of supernatant were purified in a single step using 20 mL of a Ni2+-NTA affinity column. Pure RBD was eluted using imidazol as a competitor, and the protein was dialyzed with Tris 20 mM, NaCl 150 mM, pH 7.4. The protein analysis resulted identical to the one already characterized for the production at a smaller scale fermentation, as UV-spectrum, mass spectrum, SDS-PAGE and circular dichroism were indistinguishable from those previously reported by our group [13, 14]. Protein of batches of 7L reported here were also obtained at more than 95% purity as determined by HPLC analysis 

-Line 529. In the discussion section, the authors referenced two articles (refs 79 and 80). These articles talked about RBD of SARS-CoV not SARS-CoV-2. Please update accordingly.

The information in line 559( previously line 529) was updated according to the reviewer's correction. 

-On the same note, Chen et al. (Protein Expr Purif.  2022; https://pubmed.ncbi.nlm.nih.gov/34688919/) reported that RBD could be expressed at the level of 490 mg/L of culture broth for only 3 days of induction. The operation time was shorter yet the production yield was similar. Similarly, Lee et al. (Appl Microbiol Biotechnol. 2021, https://pubmed.ncbi.nlm.nih.gov/33959781/) also reported that with the addition of glycerol fed-batch, the raw yield of RBD could reach 533 mg/L of fermentation supernatant in a 1L bioreactor for 70-hour induction. These proteins seem to be more homogenous or at least did not seem to show severe hyperglycosylation as what authors have pointed out. That being said, could the authors please justify why their production process is superior to the earlier publications?

The process referred by Chen et al (2022) is quite different from the one reported in our work. They kept DO% above 30%,  and also used two linear methanol feeding gradients for adaptation and induction. These authors did not inform if they used air of pure O2 to provide oxygenation to the culture.  In the case of Lee et al (2021), a glycerol fed-batch was made using constant feeding. No details about the induction phase feeding strategy are provided nor information regarding the oxygenation method. Our strategy uses only compressed air as the oxygen source and a DO-stat strategy, which results in a robust method that minimizes heat production and the risk of methanol accumulation while moderating oxygen requirements. The proteins reported by these authors showed less glycosylation heterogeneity because these authors mutated one of the glycosylation sites, which also reduced the expression levels due to a less solubility during protein folding and less involvement in endoplasmic reticulum quality control of glycoprotein folding.

- Lines 573-574. Due to hyperglycosylation, a commonly seen phenomenon in yeast-expressed heterologous protein, authors indicated that it could cause drawbacks for standardization and regulations compliance. Thus, further ongoing improvement was conducted toward protein homogeneity. Could authors briefly discuss the improvement plan? e.g., change different Pichia strains?

We thank the reviewer for the opportunity of clarifying this point. We have developed a method to improve protein homogeneity that is going to be published soon (manuscript in preparation). Anyway, our previous work reported that even though the glycosylation status was heterogeneous, the protein behaves identically to that expressed in mammalian cells (Sci Rep, 2020). To point out the usefulness of the preparation  even though it it heavily glycosylated it now states in the discussion section (lines 603-607):

…”Even though the glycosylation status was heterogeneous, the protein behaves identically to RBD expressed in mammalian HEK-293T cells both in conformation, stability, and immunogenic properties. Moreover, antibodies raised when using our P. pastoris produced RBD cross reacted with RBD produced in mammalian cells and vice versa [14]”....

Round 2

Reviewer 1 Report

Many thanks to the authors for providing the responses to the comments and revising the manuscript. Overall, most of the edits in the manuscript are appropriate. Some minor follow-up or additional comments are listed below to address in this revised manuscript:

1.       (Prior comment) Lines 170-172. Authors used densitometry to evaluate the RBD content in culture broth and purified fractions. Did authors use any protein standard for such quantification? (prior authors’ response) Yes indeed, the quantification was performed by comparing the densitometry with known standards of BSA.

(follow-up comment) Since authors already have purified SARS-CoV-2 RBD (ref 14), it is recommended that authors use the purified RBD for quantification purposes. Please note, in Chen et al, 2017 (https://www.ncbi.nlm.nih.gov/pmc/articles/PMC5612335/) authors mentioned that the Coomassie blue dye binds 1.33 fold less to SARS-CoV RBD than to BSA, suggesting that using BSA may not be able to provide the most accurate results.

2.       (Prior comment) Line 529. In the discussion section, the authors referenced two articles (refs 79 and 80). These articles talked about RBD of SARS-CoV not SARS-CoV-2. Please update accordingly. (prior authors’ response) The information in line 559( previously line 529) was updated according to the reviewer's correction. 

(follow-up comment) Line 559 Please note that refs 81 and 82 (originally 79 and 80) were about SARS RBD, but not about the process. Maybe the authors were referring to this manuscript: https://www.ncbi.nlm.nih.gov/pmc/articles/PMC5612335/? Please take a look and update the text accordingly.

3.       In Lines 255, 256 and 481, please correct “imidazol” to “imidazole”.

Author Response

Many thanks to the authors for providing the responses to the comments and revising the manuscript. Overall, most of the edits in the manuscript are appropriate. Some minor follow-up or additional comments are listed below to address in this revised manuscript:

  1. (Prior comment) Lines 170-172. Authors used densitometry to evaluate the RBD content in culture broth and purified fractions. Did authors use any protein standard for such quantification? (prior authors’ response) Yes indeed, the quantification was performed by comparing the densitometry with known standards of BSA.

(follow-up comment) Since authors already have purified SARS-CoV-2 RBD (ref 14), it is recommended that authors use the purified RBD for quantification purposes. Please note, in Chen et al, 2017 (https://www.ncbi.nlm.nih.gov/pmc/articles/PMC5612335/) authors mentioned that the Coomassie blue dye binds 1.33 fold less to SARS-CoV RBD than to BSA, suggesting that using BSA may not be able to provide the most accurate results.

The reviewer is correct. The use of BSA is an approximation that may produce some bias, although the same one for all samples. Still, it is extensively used as a protein quantification standard. The article mentioned by Chen et al compared BSA with RBD from SARS-Co-V and not SARS-Co-V2 so the same correction factor may not be applicable. In our case, we preferred to use BSA and not pure RBD as a standard for comparison due to the size dispersion produced by glycosylation in RBD. To make this clearer for the reader it now states in the article (lines 172-176):

“…. Chen et al (2017) [57] reported that Coommasie Blue dye binds 1.33 fold less to RBD form SARS-CoV than to BSA. It is worth mentioning that although in this work we have not compared the binding capability of RBD from SARS-CoV2 to Coommasie Blue with that of BSA, and thus there may be some underestimation of the yield of the process, this will be the same in all samples…”

  1. (Prior comment) Line 529. In the discussion section, the authors referenced two articles (refs 79 and 80). These articles talked about RBD of SARS-CoV not SARS-CoV-2. Please update accordingly. (prior authors’ response) The information in line 559 (previously line 529) was updated according to the reviewer's correction. 

(follow-up comment) Line 559 Please note that refs 81 and 82 (originally 79 and 80) were about SARS RBD, but not about the process. Maybe the authors were referring to this manuscript:

 https://www.ncbi.nlm.nih.gov/pmc/articles/PMC5612335/? Please take a look and update the text accordingly.

We apologize by the mistake. Reference in line 566 have now been updated to Chen et al 2017 (doi: 10.1016/j.xphs.2017.04.037).

  1. In Lines 255, 256 and 481, please correct “imidazol” to “imidazole”.

We thank the reviewer for detecting the mistakes. They have been now corrected.

Reviewer 3 Report

All necessary corrections were made. The paper can be accepted for publication in the journal Fermentation.

Author Response

Thanks a lot for your suggestions

Round 3

Reviewer 3 Report

Paper can be accepted.

Author Response

Thanks a lot for your suggestions